# Genomic Context of SARS-CoV-2 Outbreaks in Farmed Mink in Spain during Pandemic: Unveiling Host Adaptation Mechanisms

**DOI:** 10.3390/ijms25105499

**Published:** 2024-05-17

**Authors:** María Iglesias-Caballero, Vicente Mas, Sonia Vázquez-Morón, Mónica Vázquez, Sara Camarero-Serrano, Olga Cano, Concepción Palomo, María José Ruano, Cristina Cano-Gómez, José Antonio Infantes-Lorenzo, Albert Campoy, Montserrat Agüero, Francisco Pozo, Inmaculada Casas

**Affiliations:** 1Reference and Research Laboratory for Respiratory Virus, National Centre for Microbiology, Instituto de Salud Carlos III (ISCIII), 28220 Majadahonda, Madrid, Spain; vmas@isciii.es (V.M.); svazquez@isciii.es (S.V.-M.); pacopozo@isciii.es (F.P.); 2CIBER de Epidemiología y Salud Pública (CIBERESP), Instituto de Salud Carlos III (ISCIII), 28029 Madrid, Spain; 3Central Laboratory of Veterinarian (LCV), Ministry of Agriculture, Fisheries and Food, 28110 Algete, Madrid, Spain; mruanor@mapa.es (M.J.R.); ccano@mapa.es (C.C.-G.); maguerog@mapa.es (M.A.); 4CIBER de Enfermedades Respiratorias (CIBERES), Instituto de Salud Carlos III (ISCIII), 28029 Madrid, Spain

**Keywords:** SARS-CoV-2, minks, NGS

## Abstract

Severe acute respiratory syndrome coronavirus 2 (SARS-CoV-2) infects various mammalian species, with farmed minks experiencing the highest number of outbreaks. In Spain, we analyzed 67 whole genome sequences and eight spike sequences from 18 outbreaks, identifying four distinct lineages: B.1, B.1.177, B.1.1.7, and AY.98.1. The potential risk of transmission to humans raises crucial questions about mutation accumulation and its impact on viral fitness. Sequencing revealed numerous not-lineage-defining mutations, suggesting a cumulative mutation process during the outbreaks. We observed that the outbreaks were predominantly associated with different groups of mutations rather than specific lineages. This clustering pattern by the outbreaks could be attributed to the rapid accumulation of mutations, particularly in the ORF1a polyprotein and in the spike protein. Notably, the mutations G37E in NSP9, a potential host marker, and S486L in NSP13 were detected. Spike protein mutations may enhance SARS-CoV-2 adaptability by influencing trimer stability and binding to mink receptors. These findings provide valuable insights into mink coronavirus genetics, highlighting both host markers and viral transmission dynamics within communities.

## 1. Introduction

Severe acute respiratory syndrome coronavirus 2 (SARS-CoV-2) has the ability to infect multiple mammalian host species as other Betacoronaviruses [1]. SARS-CoV-2 shows a broad host tropism infecting many species due to the potential affinity of viral spikes for animal angiotensin-converting enzyme 2 (ACE2) orthologs [2,3]. In addition, coronaviruses are capable of adapting to new environments through mutation and recombination which provides the potential for host range and tissue tropism changes [4]. The multiple sequence alignments of the ACE2 proteins show high similarity among humans, dogs, cats, tigers, minks, and other animals [5]. Reverse zoonotic events were described as a consequence of the viral introduction from farm workers into densely populated farms, these living conditions of farmed minks could lead to rapid transmission between animals and subsequently to humans [6,7].

The coronavirus spike protein is responsible for the viral entry into the host cell. During viral infection, the spike glycoprotein is cleaved into S1 and S2 subunits, where S1 contains the spike receptor-binding site (RBD), which directly binds to the region located in the peptidase domain of the ACE2, and S2 subunits are in charge of membrane fusion [8,9]. A series of amino acid positions are crucial for molecular interactions between the RBD and the interacting surface of ACE2 [10]. Previous works proposed that a modification of the RBD structure is needed for SARS-CoV-2 to bind to the ACE2 [11]. 

The analysis of different farm outbreaks revealed the recurrent detection of several mutations located in the RBD of the spike protein after circulation in mink: Y453F, F486L, and N501T [12,13]. Moreover, the Y453F and the N501T substitutions have also been associated with an increment of the affinity to the human ACE2 receptor [14]. A similar mutation in the position 501 (N501Y) is present in several SARS-CoV-2 variants of concern (VOCs) showing it being hypothetically responsible for high transmission [15,16,17]. RBD is a mutation hotspot in the genome of SARS-CoV-2 and the appearance of mutations can affect the biophysical characteristics of the protein with great alterations in binding affinity to ACE2 because of the change in only 2–4 amino acids [18].

The first SARS-CoV-2 outbreaks in farmed mink caused by human-to-animal spillover events occurred in the Netherlands in April 2020 [19]. Later, in November 2020, an outbreak in Denmark [6,20] raised significant concerns regarding the emergence of a mink-associated SARS-CoV-2 lineage circulating in humans and minks in farms located in Northern Jutland [18,19]. This “mink-derived” lineage, known as the “cluster-5 variant”, was concerning due to its rapid transmission and the possession of five mutations occurring at relevant positions within the spike: H69/V70 deletion, Y453F, D614G, I692V, and M1229I. These changes were worrisome as they raised the possibility of a partial viral immune escape [21,22,23], especially at a time when the vaccination was unavailable in Europe and herd immunity remained fragile.

Active surveillance in mink farms has been instituted across 15 countries, Spain included [24,25], yielding positive outcomes in terms of outbreak detection and control. However, challenges persist stemming from the difficulty in identifying transmissions prior to symptom onset and the potential for unnoticed spread [26]. Enhancing data collection efforts could lead to improvements in surveillance systems and public health interventions.

In this study, we performed a molecular epidemiology analysis of 18 outbreaks in mink farms in Spain, with a specific focus on the genomic features that provide insights into the evolution, genetic host markers, and community distribution. Moreover, a phenotypic investigation of the spike proteins originating from these minks can offer clarity on the potential implications of the observed genomic changes and provide insights into the biological properties of these spikes.

## 2. Results

### 2.1. Spanish Mink Sequences

We analyzed 117 nucleic acid extracts from 12 different outbreaks in farmed mink, resulting in 68 whole genome sequences. Additionally, we obtained eight partial spike sequences from samples that could not undergo full sequencing. The cycle-threshold values and the sample quality were crucial factors in achieving complete genome sequences, serving as key methodological checkpoints in our sequencing workflow (Appendix A). Among the sequences meeting these quality criteria (56 out of 68), the average depth of coverage ranged from 318× to 14,381×. For samples with genome coverage between 70 and 90%, or those with satisfactory cycle-threshold values but incomplete sequencing results for the spike gene by NGS, we re-amplified and sequenced genomic fragments covering the spike gene using the Sanger sequencing method. 

Lineage assignment revealed the presence of four different lineages (Table 1): B.1, B.1.177, B.1.1.7, and AY.98.1. All sequences belonging to B.1 lineage originated from the same outbreak (Table 1), which was the largest outbreak analyzed in Spain. The sequence belonging to the AY.98.1 lineage was the sole complete sequence retrieved from that outbreak (Table 1), posing challenges for studying delta viruses in mink infections. In contrast, sequences from the B.1.177 and B.1.1.7 lineages were sourced from diverse outbreaks, offering a comparative framework across sequences from various farms within the same lineage. The genomes obtained in this study significantly increased the pool of mink available sequences for the B.1.1.7 and AY.98 lineages in public genomic databases.

### 2.2. Genomic Context

To characterize the genomic features of the Spanish SARS-CoV-2 minks’ sequences, we conducted a phylogenetic analysis to establish the relationship between the studied viruses and other coronaviruses found in minks. Additionally, we identified not-lineage-defining mutations. Initially, we conducted a phylogenetic analysis using high-quality complete genome mink sequences available in the GISAID database, alongside the Spanish sequences obtained in this study. This analysis revealed that the association of mink sequences primarily occurred based on the outbreak events rather than the lineage classification, particularly within ancient lineages (Figure 1). Upon further exploration of the clusters, it became apparent that different groups were predominantly linked by the outbreak occurrences rather than the lineage classifications, potentially attributable to the accumulation of mutations associated with these events (Figure 2).

When specifically examining the B.1 lineage sequences, the association with outbreaks became notably clear, delineating ten different B.1 clusters with strong support (Figure 2). The differentiation of these groups was driven by the cumulative changes observed in the ORF1ab polyprotein, spike, and nucleocapsid genes when compared to the reference sequence Wuhan-Hu-1 (GenBank accession: NC_045512) (Figure 2). Notably, the polyprotein ORF1a exhibited the highest number of changes among the analyzed proteins, thus playing a pivotal role in defining different clusters even when the spike sequences were identical. Recurrent changes were observed among the B.1 clusters, particularly in the spike and in the ORF1a polyprotein. The mutations T256I and G4177E were the most frequently detected, present in five different clusters (Figure 2). These recurrent ORF1a mutations, coupled with other changes in the polyprotein and spike, contributed to the formation of distinct groups. Interestingly, the spike mutation N501T was identified in seven distinct clusters. These clusters (clusters 2–6 and sub-clusters 8a and 8b as shown in Figure 2) exhibited the N501T mutation either alone or in combination with other mutations, such as G142D and F486V/L in the spike gene. Despite the recurrence of several mutations, the unique combination of mutations was associated with specific outbreak events and could be easily traced within the population. 

Among the Spanish sequences, a highly supported cluster (97.4 bootstrap) was identified, further comprising two well-defined sub-clusters. This cluster had three shared ORF1a mutations, W2008C, L3199S, and G4177E. However, the two sub-clusters within this cluster demonstrated distinct spike protein variants (sub-clusters 8b and 8a in Figure 2). One sub-cluster contained the N501T mutation in the spike protein, while the other sub-cluster featured the TVH combination (N501T, F486V, and D796H). Notably, the cluster 8a also possessed an additional mutation, G2640R, in the ORF1a polyprotein. 

The B.1.177 lineage cluster exclusively comprised the Spanish sequences, whereas the remaining sequences belonged to the B.1.177.60 sub-lineage, forming a separate cluster (Figure 3, top panel). All the sequences from this sub-lineage originated from the same source in Latvia, underscoring the close relationship between viruses, outbreaks, and geographical regions. Within the well-supported B.1.177 Spanish cluster, three distinct sub-clusters were identified, primarily characterized by not-defining mutations found mainly in ORF1a and spike genes. Sub-cluster 1 exhibited the mutation K776R in the spike protein, sub-cluster 2 presented an additional mutation, P3952S, in the ORF1a polyprotein, and sub-cluster 3 consisted of sequences with the mutations A656V in the ORF1a polyprotein and Y453F in the spike protein. Each sub-cluster corresponded to a different outbreak event (Figure 3). As for the mink sequences associated with the B.1.1.7 lineage, only one sequence had been previously published from Poland (hCoV-19/mink/Poland/53970/2021, EPI_ISL_9640055). The phylogenetic analysis indicated that this sequence clustered with Spanish sequences but formed a different sub-group defined by the presence of the Y453F mutation in the Polish minks. Comparatively, the Spanish sequences exhibited similarity to the alpha reference sequence (hCoV-19/England/MILK-9E05B3/2020, EPI_ISL_601443), characterized by a combination of three not-defining mutations in the N protein, G5V, A182S, and S235F (Figure 3, bottom panel).

The Spanish sequence associated with the AY.98.1 lineage (hCoV-19/mink/Spain/R2990-21-1) was the sole published mink sequence belonging to this lineage (Figure 1). In comparison with other AY.98.1 sequences, the only additional mutation identified was R158G in the spike protein. Although this mutation has also been observed in other delta variant sequences, its global prevalence remains relatively low.

The presence of a significant number of not-lineage-defining mutations closely associated with different outbreaks suggests that the virus is undergoing constant genetic changes as it adapts to its hosts. This observation, combined with the occurrence of recurrent mutations across various lineages, underscores the evolutionary struggle of the virus in adapting to different hosts. The phylogenetic analysis conducted using complete genome sequences revealed the emergence of distinct mutations that effectively define nearly every individual outbreak. These specific changes provide valuable insights into tracking the impact of these outbreaks, not only within minks but also within the human population.

### 2.3. High Mutation Rate in Non-Structural Proteins

Comparing the ORF1ab polyprotein of the Spanish minks to other protein sequences revealed a significant number of additional mutations beyond the lineage-defining ones. Particularly, the replicase polyprotein 1a region displayed the highest mutation count across different lineages and outbreaks. A majority of these mutations were concentrated in the multidomain NSP3 protein (Figure 4B). These mutations span multiple protein domains, including the PLpro (S830P and T976A), whose activity has been suggested as indicative of the virus’s host species compatibility [27]. Several mutations were consistently present across different outbreaks, notably W1190C, Q1821R, and G1822R in the membrane-associated regions of NSP3, along with L436S at the C-terminal region of NSP4. Such mutations, located at the cytosolic side of membrane-associated proteins, may modulate interactions between host factors or viral NSPs, influencing the formation of internal double-membrane vesicles where viral replication and transcription occur. Additionally, in the sequences from the 2020/1 outbreak (B.1 lineage), two changes were detected in the replication transcription complex (RTC) (Figure 4C). These changes included the mutations G37E in NSP9 and S486L in NSP13, and the primary mutation detected in the ORF1b region (Figure 4B). The mutation G37E in NSP9 was also detected in the B.1.177 lineage outbreaks.

An analysis of mutation frequency within the ORF1ab, based on the available sequences in GISAID, revealed a significant finding regarding the Spanish mink sequences. Among all mutations identified in the Spanish mink sequences, the G37E in NSP9 is strongly associated with a host marker specifically found in minks. This mutation, present in various lineages, was detected in 23% of the mink sequences available in GISAID. In contrast, it was significantly more prevalent in the Spanish mink sequences, accounting for 63% of those (Figure 4B). In contrast, the mutation L436S, also identified in the lineages B.1, B.1.1.7, and B.1.177, was present in 75% of the Spanish sequences but exhibited a very low occurrence in mink samples from other regions of the world. Similarly, other mutations in NSP3 (W1190C, G1822R) and in NPS13 (S468L) identified in our study had limited representation in the GISAID database (Figure 4B). 

### 2.4. Spike Sequences Exhibit Host Adaptive Mutations

In the Spanish mink sequences, not-lineage-defining mutations detected in the spike protein spanned various domains of the spike S1 and S2 subunits, predominantly observed in the lineages B.1 and B.1.177 (Figure 5B). Regarding the S1 subunit, three of these mutations were in the RBD (Y453F, F486V, and N501T). Additionally, three other mutations were identified at the NTD domain (L54F, R158G, and V227I), and one at the SD1 domain (D571G). It is important to note that different changes at RBD positions 453, 486, and 501 have been previously associated with adaptive mechanisms in minks, improving RBD binding properties to the mink ACE2 receptor [28,29,30]. Notably, in our study, we identified a novel, previously undescribed combination of these RBD changes (VTH) where valine and threonine are present at residues 486 and 501, respectively. In the S2 subunit, several mutations were identified in inter-protomer regions (F759S, K776R, P793S, and D796H). In addition, a change was detected at the HR1 domain (S967N), a critical region that refolds during the spike transition from prefusion to postfusion conformation during the membrane fusion process. These mutations in the S2 subunit suggest possible adaptative changes related to spike prefusion trimer stability. Moreover, a rare change at the cleavage site S2 (S816T) was found in only a few mink sequences. This mutation may play a role as an adaptive change, potentially facilitating proper spike priming at S2 by host cellular proteases. 

Considering all sequences available in GISAID, the changes at RBD residues (453, 486, and 501), detected across different lineages, appear to be the ones most strongly associated with host markers in minks. The substitutions of phenylalanine 486 to leucine, isoleucine, or valine are present in 65% of the mink sequences. For Y453F and N501T, the percentages reach the 40% and 80%, respectively. Although the K776R mutation is relevant in the Spanish mink sequences of the lineage B.1.177 (14%), it is less commonly represented when considering all the available sequences in GISAID (Figure 5B). This suggests that the K776R mutation is not as widespread in mink populations beyond the Spanish lineage B.1.177 sequences.

To investigate whether certain spike protein changes detected in the Spanish mink viruses were associated with host adaption mechanisms, we conducted SPR experiments to assess the impact of RBD spike mutations on their binding to the mink ACE2 receptor. For this purpose, we generated and expressed six different HexaPro spike constructs based on the mink sequences. Four of the HexaPro constructs were derived from the most predominant mink virus sequences detected in Spanish farms, including B.1-N501T, B.1.177-Y453F, B.1-VTH, and a partial N501T/F486V construct. The other two constructs represented mink sequences available in GISAID, B.1-N501T/F486L and the cluster V-like construct B.1Δ69-70/Y453F. Additionally, we produced and tested the HexaPro constructs of human lineages B.1, B.1.177, and B.1.1.7 as controls for direct comparison with the spikes derived from mink viruses. 

As described in previous studies focusing mainly on RBD fragments [29], we found that the interaction between the B.1 spike and the monomeric form of the mink ACE2 was notably weak (Figure 6 and Appendix A). Only by increasing the mink ACE2 concentration up to 50 µM, we were able to detect a significant level of binding with the B.1 spike. Working at this mink ACE2 concentration, the level of binding was markedly enhanced with all mink spikes (Figure 6 and Appendix A), particularly in the cases containing the Y453F mutation (cluster V and B.1.177-Y453F) or the double mutations at the residues 486 and 501 (B.1-N501T/F486L, B.1-N501T/F486V, and B.1-VTH). In the B.1 lineage, the single mutation N501T also exhibited a significant improvement in binding to mink ACE2, approximately half of the total observed with the double mutants at the positions 486 and 501 (Figure 6). Among the human-derived spikes, the alpha spike showed the strongest binding level to the mink ACE2. All these data support the notion that the mink mutations associated with the RBD domain are of an adaptive nature, enhancing the interaction between the spike and mink ACE-2, which may have implications for host adaptation.

### 2.5. Mutations Detected in Mink Variants May Have the Potential to Trigger Spillover Events from Minks to Humans

In our study, we aimed to assess the potential transmission of mink variants to humans by evaluating the affinity and kinetics of the interaction between the mink spikes and the human cell receptor ACE2 (hACE2). Through SPR experiments, we observed that all constructs containing mink mutations efficiently bind to the hACE2. Compared to the spike of human ancient viruses (B.1), constructs containing the mutation Y453F (mink cluster V and B.1.177-Y453F) or the single mutation N501T exhibited significant improvements in affinity, indicated by lower KD values, for the cellular receptor (Table 2 and Appendix A). These enhancements in affinity were primarily characterized by a higher stability of spike–receptor complexes with 5–10 times slower koff rates. Additionally, the B.1-N501T spike demonstrated a slightly faster association for hACE2. Conversely, constructs containing double mutations at positions 486 and 501, either with the combination leucine/threonine (B.1-N501T/F486L) or with the combination valine/threonine (B.1-N501T/F486V and B.1-VTH), displayed lower affinity for the hACE2 compared to the B.1-N501T spike (Table 2 and Appendix A). These findings suggest that the adaptive changes observed in the RBD of mink variants may potentially allow spillover events from minks to humans.

## 3. Discussion

It is well known that many animal species can be infected with emerging coronaviruses. However, the potential risk of emerging coronavirus transmissions to humans particularly from infected farmed minks raises crucial questions about mutation accumulation and its impact on viral fitness, transmission dynamics within these facilities, and the efficacy of vaccines or treatments against these animal viruses in the human population. To face this risk, the association of animal and human health laboratories in this work, under the One Health umbrella, enables the characterization of mink viruses with all the technology developed during the pandemic for the analysis of human SARS-CoV-2, enhancing virus surveillance by moving beyond mere sequence description in mink’s coronaviruses.

In mink farms, the close living conditions, with cages placed adjacent to each other, facilitate contagion, allowing nose-to-nose contact and air exchange among animals. Previous studies have shown that air, fur, and straw samples from mink farms can test PCR-positive for SARS-CoV-2 [19,31], indicating the possibility of indirect transmission through feed, bedding, and airborne routes within the farms [32]. Moreover, SARS-CoV-2 infections in farmed minks may go unnoticed initially. Boklund et al. reported that about 30% of the interviewed farmers in Denmark had not observed symptoms in the animals while the infection was already spreading within the farm [31]. Similarly, Hammer et al. [6] described a rapid spread of the virus with a significant increase in viral detection from 13% to 86% within just four days. In Spain, during 2020–2021, 18 outbreaks were declared in mink farms across four distinct regions of the country. Interestingly, 15 out of these 18 outbreaks occurred in farms located in Galicia, the main producer of mink fur in Spain [33]. The combination of high and rapid transmission rates, along with the lack of clinical symptoms in some cases, may have contributed to the emergence of a substantial number of mutations that could enhance viral adaptation to minks before the outbreak could be effectively controlled. This emphasizes the critical need for surveillance and containment measures to prevent the spread of SARS-CoV-2 in mink farms and to closely monitor potential risks of transmission to humans. 

The genomic analysis conducted in this study involved high-quality sequences obtained from infected minks during the mentioned outbreaks in Spain. The sequences obtained represented four different lineages: B.1, B.1.177, B.1.1.7, and AY.98.1. Interestingly, although these lineages were coincident with those circulating in humans at the time, they exhibited new mutations not present in circulating human SARS-CoV-2. 

The phylogenetic tree analysis of all SARS-CoV-2 mink sequences from Spain provided valuable insights. Surprisingly, the sequences did not group according to their respective lineages, as typically observed in viral phylogenetic trees. Instead, the sequences clustered together based on the specific outbreaks they were associated with. This clustering pattern by outbreak is significant and may be attributed to the rapid accumulation of mutations, particularly in the ORF1a polyprotein and in the spike protein. These mutations could potentially be linked to the fast transmission of the virus within the farmed minks. Additionally, the living conditions of the animals, their proximity to each other in cages, and their isolation from the community might have contributed to this unique pattern of viral evolution. This finding highlights the dynamic nature of viral evolution in mink populations and underscores the importance of understanding how specific outbreaks can lead to the emergence of distinct viral variants. The fast accumulation of mutations in mink-associated sequences raises concerns about the potential risk of spillover events from minks to humans. 

In a previous study on genomic diversity and recurrent mutations, both the ORF1ab and the spike genes were found to exhibit a particularly large number of recurrent mutations. This study raised interest in investigating specific locations and proposed the NSP6, NSP11, and NSP13 in the polyprotein where recurrent mutations occurred, indicating the viral evolution and adaptation of SARS-CoV-2 to the human host [34]. Additionally, the NSP3 protein emerged as another important candidate among those proteins of interest. The Spanish mink SARS-CoV-2 sequences displayed a high number of mutations in the NSP3 protein across different lineages. Some of these mutations were located in the critical PLpro domain, a region of notable importance and a promising target for developing a new class of antivirals for coronaviruses [35]. Research has suggested that deISGylase activity mediated by viral protease PLpro may indicate the species that SARS-CoV-2 can infect [27]. In the context of mink infections, mutations in the finger (S830P) and thumb (T976A) subdomains of the PLpro were detected during the outbreak in 2020/1 (B.1 lineage). These mutations harbor the potential to alter substrate preferences, thereby modulating deubiquitinating or deISGylating activities. Consequently, the virus could enhance its ability to evade mink antiviral immunity. Investigating the effects of these specific mutations in the PLpro domain is of significant interest, as it offers insights into the potential mechanisms by which SARS-CoV-2 adapts to and interacts with minks.

The mutation G37E in the NSP9 protein is a primary candidate for consideration as a host marker in minks, given its identification in various mink outbreaks of different lineages worldwide, including those studied in Spain. Our findings align with previous studies that have also identified this mutation as a potential host marker [12]. The high prevalence of the G37E mutation among the mink SARS-CoV-2 sequences, coupled with its low prevalence in viruses from the human population, supports the hypothesis that it may play a significant role in mink host adaptation. Moreover, the presence of both glutamic acid (E) and arginine (R) at the position 37 of NSP9 in mink sequences underscores the evolutionary relevance of this residue. The detection of the G37E mutation in NSP9, in conjunction with the S486L mutation in NSP13, in the sequences from the outbreaks in 2020/1 and 2021/1 in Spain is of particular interest. Both residues are located in surface grooves directly involved in RNA binding [3,36,37]. Given the non-conservative nature of both changes, these mutations could significantly impact the stability of NSP-RNA complexes, ultimately modulating the proposed functions of both proteins in the viral RNA transcription complex (RTC). Regarding NSP9, it is suggested to play a role in the intermediate formation during the mRNA capping [36]. The G37E mutation could potentially alter the stability of NSP-RNA interactions during this process, affecting mRNA capping efficiency. Additionally, residue 37 is located in a contact loop involved in NSP9 multimerization [38]. The oligomerization state of NSP9 influences its nucleic acid-binding function and its ability to interact with the NSP12 RNA-dependent RNA polymerase in the RTC [36,38]. Similarly, NSP13 is involved in RNA template accommodation during RNA elongation/backtracking [39,40]. The S486L mutation, in combination with the G37E mutation in NSP9, could affect the stability of the NSP-RNA complex during RNA elongation, potentially modulating the RNA synthesis efficiency and accuracy. 

The spike protein in SARS-CoV2 is known to have a high variability and serves as a key target for host adaptation. In the mink sequences, several amino acid changes were observed that may act as adaptive mechanisms similar to those seen in other hosts. These mutations are located at inter-protomeric regions, in the HR1 domain, and in the S2 cleavage site. S2 mutations could potentially modulate trimer stability or priming, similar to the adaptive mechanisms described in other surface glycoproteins of respiratory viruses such as the hemagglutinin in avian influenza viruses, during adaptation to humans [41]. These modifications in the spike protein may enhance viral entry and replication in mink cells, contributing to increased viral fitness in minks. Moreover, our study provided evidence of adaptive mechanisms based on an improvement of the spike binding to mink ACE2, the cellular receptor used by SARS-CoV-2 to enter host cells. Specifically, we identified host markers related to changes at the RBD positions 453, 486, and 501. These mutations could potentially enhance the affinity of the spike protein for the mink ACE2 receptor, facilitating viral entry and replication in minks. The mutations Y453F and N501T, detected in Spain in sequences from the lineages B.1.177 and B.1, respectively, have been previously shown to strongly improve spike binding to mink ACE2 [28,29]. The Y453F mutation initially was identified as part of the “cluster 5” which raised international concern due to its potential impact on neutralization at a time when vaccines or treatments were not yet available. A preliminary report suggested reduced neutralization titers against this cluster with low and intermediate titers [21]. However, subsequent studies clarified that Y453F does not affect neutralizing antibodies [23] but does improve spike and hACE2 interaction [14]. Despite its prevalence in mink sequences, the Y453F mutation has not been commonly detected in human lineages circulating in the community, as it was defined in approximately 2625 viruses of 31 different lineages but has not been classified as part of any circulating variant of concern. The lack of this mutation in the human lineages could be explained by the fact that while it may confer benefits in terms of binding to hACE-2, its impact on transmission in humans may be limited. As mentioned earlier, the improvement in binding to hACE2 achieved by the Y453F mutation is primarily characterized by an increased stability of RBD-receptor complexes. This specific change might not confer a significant evolutionary advantage for transmission in humans [12], as evidenced by in vitro studies that Y453F attenuates SARS-CoV-2 in human bronchial cells [13]. On the contrary, the mutation N501T has been shown to improve the spike association kinetics to hACE-2, making it easier for the virus to attach to host cells and potentially facilitating transmission in humans. This mutation N501T was first detected in humans in the early stages of the pandemic, before being detected in SARS-CoV-2 outbreaks in their respective animal hosts [12]. Additionally, the characterization of this change in other animal hosts, such as white-tailed deer, indicates a potential relation among this change and animal infection [42].

Mutation at the position 486 in the spike protein is indeed a crucial determinant in mink sequences, with the potential for up to three different amino acid substitutions: leucine, valine, and isoleucine. Among these changes, the mutation F486L is the most frequently detected in minks, and the previous analyses of Dutch farm outbreaks have shown that viruses carrying this mutation may evolve and transmit at an accelerated rate in minks [43]. This enhancement in transmission is further supported by various studies including our own data, which demonstrate that leucine at the position 486 enhances binding to mink ACE2 [29]. Moreover, SARS-CoV-2 pseudoviruses carrying the F486L mutation have been shown to gain the ability to infect HeLa cells by overexpressing the mink receptor [30]. These findings underscore the significant role of the F486L mutation in facilitating viral transmission in minks. Similarly, the RBD mutation F486V present in the VTH sequences from the Spanish farms may also contribute to viral transmission in minks by improving spike binding to the mink cellular receptor. Notably, the emergence of lineage-defining mutations at the position 486, including substitutions to valine, serine, or proline, in the Omicron variant raises questions about the susceptibility of minks to this variant. To address this concern, further functional studies are required. Our data on spike protein binding to human ACE2 indicates that viruses identified in minks have the potential to trigger spillover events from minks to humans. However, the reduced number of changes observed in the spike protein regions targeted by human neutralizing antibodies suggests that such potential transmission may be limited in a human population with adequate immune status. Nevertheless, given the location of some changes, mink-derived viruses may exhibit reduced sensitivity to certain treatments for humans. For instance, the F486V mutation has been associated with a lower susceptibility to already approved treatments based on antibodies targeting the spike RBD, such as Ly-Cov555, AZD8895, or REGN10933 antibodies [44,45,46]. Additionally, mutations at PLpro, originating from SARS-CoV-2 adaptation to minks, as identified in this study, may also result in a decreased viral susceptibility to novel antivirals designed to target the papain-like protease, which are currently under development [47]. This underscores the importance of ongoing surveillance and vigilance to monitor the dynamics of these mink variants and their potential implications for human health. It is crucial to obtain the complete sequence of the viral genome, rather than only analyzing the spike protein, to gain more comprehensive phylogenetic information about viral sequences, leading to more accurate clustering. Furthermore, analyzing the polyprotein in conjunction with the spike can provide a more thorough understanding of host markers and viral transmission in the community, thereby enhancing our knowledge of various outbreaks. 

This work demonstrates the potential of the development of methodology in the surveillance of human SARS-CoV-2 viruses that could be used for monitoring animal coronaviruses. Our study highlights the importance of conducting phylogenies with complete genomes, as it underscores the significance of characterizing the entire genome for tracing outbreaks or mechanisms of host adaptation. Additionally, this work illustrates the importance of phenotypic virus studies, not considering only the hypothetical impact of genetic changes, showing that only through the combination of genomic and functional/antigenic methods can surveillance networks propose appropriate responses to public health emergencies.

## 4. Materials and Methods

### 4.1. Sample Origin and Preparation

From June 2020 to June 2021, the Central Laboratory of Veterinarian (LCV) analyzed 8.867 samples originating from the 29 actively operating mink farms across Spain during that period. These farms were geographically distributed as follows: 26 in Galicia, 1 in Aragón, 1 in Valencia, and 1 in Castilla y León. The samples were collected via oropharyngeal swabs and preserved in a virus-inactivating medium. Subsequently, nucleic acids were extracted from these samples using the commercial kit IndiMag Pathogen kit (Indical, San Francisco, CA, USA) in a BioSprint 96 automated extraction system (Qiagen, Hilden, Germany). 

### 4.2. Virus Detection by RT-PCR 

The nucleic acid extracted was analyzed by two distinct real-time RT-PCR assays, each following established protocols and targeting specific regions within the virus genome. One assay was designed to detect the gene E [48], while the other was tailored to identify the gene N [49]. Both RT-PCRs included the detection of the B-actin gene as internal control and have received accreditation at the LCV in accordance with ISO 17025 standards [50] (PESIG/DM-01+IESIG/DM-117). Out of the total samples, 351 were found to be positive for SARS-CoV-2 in 18 outbreaks (1 in 2020 and 17 in 2021) through these assays. A total of 117 RNA extracts from strongly positive samples (Ct values < 25) from each outbreak were forwarded to the Reference and Research Laboratory for Respiratory Virus (CNM, ISCIII) for subsequent sequencing and genomic analysis over the course of 2020 and 2021. From the 117 samples received, 68 complete genomes were successfully recovered with a coverage greater than 90%, representing 12 out of the 18 outbreaks. Table 1 presents comprehensive information including farm identification, the date of outbreak notification to the national level, sample ID sequencing, lineage, and the GISAID data bank identification of the samples.

### 4.3. Sequencing Methods

The whole genome amplification for NGS was performed at the National Center of Microbiology (ISCIII). Sample cDNAs were obtained using a Sequence Independent, Single-Primer-Amplification (SISPA) with the random primer FR26RV-N (5′-GCCGGAGCTCTGCAGATATCNNNNNN-3′) [51]. Subsequently, cDNA products were amplified following the ARTIC network’s PCR protocol, using the primer pool version “ARTIC n-CoV-2019” v3 and v.3.5 [52]. Libraries were prepared according to the instructions of the Nextera DNA library preparation kit (Illumina, San Diego, CA, USA). For library barcoding, unique dual indexes were used, and subsequently, the libraries were pooled together. Sequencing was carried out using a NextSeq Reagent kit 300 cycles (Illumina, CA, USA) on a NextSeq sequencer (Illumina, CA, USA). In cases where the sequence of the spike gen was not complete or to confirm specific sequences of interest within the spike, two PCR assays targeting the S1 and S2 regions were conducted. For the amplification of the S1 region, 5 μL of nucleic acid extraction was added into 20 μL of QIAGEN OneStep RT-PCR Kit reaction mixture with the primers S1F (0.4 µM) (5′-GACATGAGTAAATTTCCCC-3′) and S1R (0.4 µM) (5′-GCCCCTATTAAACAGCC-3′). The amplification conditions consisted of a reverse transcription (RT) step at 50 °C for 30 min, an initial denaturation at 95 °C for 15 min, followed by 45 cycles of denaturation at 95 °C for 30 s, annealing at 60 °C for 2 min, and extension at 68 °C for 3 min. For amplifying the S2 region, the primers used were the ARTIC pool primers 77L and 84R. The amplification conditions consisted of an RT at 50 °C for 30 min, 95 °C for 5 min, followed by 45 cycles, 95 °C for 30 s, and 67 °C for 2 min. The amplified products, approximately 2490 bp for S1 and 2551 bp for S2, were visualized by performing electrophoresis on a 1% agarose gel. To obtain high-quality sequencing data, the amplified products of the expected sizes were purified and subjected to double-strand sequencing using the Sanger chain-termination method. This sequencing was carried out using the BigDye Terminator v3.1 Cycle Sequencing Kit on an ABI PRISM 3700 DNA Analyzer (Applied Biosystems, Waltham, MA, USA).

### 4.4. Quality Control of the Sequences

The quality control after sequencing involved several aspects. It included an evaluation of the number of Ns (undetermined bases) in the consensus genome, the presence of ambiguous bases (non-A, T, C, or G), the identification of mutations unique to specific consensus genome, and the detection of frameshifts and premature stop codons. To perform this quality control, the Nextclade sequence quality check tool v.3 (https://clades.nextstrain.org/, URL accessed on 12 September 2023) was employed. In addition, the mean depth of the sequence reads and the percentage of viral coverage were also considered as quality markers. These factors provided an assessment of the sequencing depth and the extent of coverage of the viral genome. These quality markers played a crucial role in determining the suitability of the sequences for further analysis.

### 4.5. Bioinformatics Analysis

Sequencing samples were analyzed for viral consensus genome reconstruction using the viralrecon pipeline (https://github.com/nf-core/viralrecon) written in Nextflow (https://www.nextflow.io/) in collaboration with the nf-core (https://nf-co.re/) community and the Bioinformatics Unit of the Instituto de Salud Carlos III (BU-ISCIII) (https://github.com/BU-ISCIII). In this pipeline, fastq files containing raw reads were first analyzed for quality using FastQC v0.11.9 (http://www.bioinformatics.babraham.ac.uk/projects/fastqc/). The raw reads were trimmed using fastp v.0.20.1 [53]. The sliding window quality filtering approach was performed, scanning the read with a 4-base-wide sliding window and cutting 3′ and 5′ ends of the base when the average quality per base dropped below a Qphred33 of 30. The reads shorter than 50 nucleotides and the reads with more than 10% of read quality under Qphred 30 were removed. Additionally, poly-X sequences were removed from read ends. The trimmed reads were mapped against the reference SARS-Cov2 genome (NC_045512.2) with bowtie2 v.2.3.5.1 [54]. Amplicon primers were then soft-clipped from mapping files using iVar v.1.2.2 [55]. Picard v.2.22.8 (https://github.com/broadinstitute/picard) and SAMtools v.1.9 [56] were used to generate viral genome mapping stats. To obtain statistics about host genome content, the kmer-based mapping of the trimmed reads against the GRCh38 NCBI human genome reference was performed with Kraken2 v.2.0.9beta [57]. Variant calling was performed using VarScan2 v.2.4.4 [58], which calls for low- and high-frequency variants from which the variants with an allele frequency higher than 80 were kept to be included in the consensus genome sequence. Both variants included and not included in the consensus genome sequence were annotated using SnpEff v.4.5COVID19 [59] and SnpSift v.4.3.1t [60]. Finally, bedtools v2.29.2 [61] was used to obtain the viral genome consensus with the filtered variants and mask genomic regions with coverage values lower than 10×. Final summary reports were created using MultiQC v.1.9 [62]. The variant-calling files (vcf) obtained with VarScan2 containing all of the variants (low and high frequency) were analyzed with Bcftools v.1.10.2 to merge all of the samples and select the allele frequency field [63].

Lineages were assigned with the command line tool of the pangolin nomenclature system (https://cov-lineages.org/resources/pangolin.html).

### 4.6. Alignment and Phylogenetic Analysis

Two different alignments were generated: one including all the SARS-CoV-2 mink’s sequences available in GISAID with good quality being the n 1113 sequences (EPI_SET ID: EPI_SET_240109zm, doi:10.55876/gis8.240109zm); the other is an alignment with all the Spanish sequences, where n is 68. The alignment was performed including the Wuhan-1 reference genome (GenBank: NC 045512) [64] using MAFFT [65]. Maximum-likelihood phylogenies were built with IQTREE software v.2.3.2 [66] using a GTR model, and a bootstrap test was replicated 1000 times. The phylogenies were rooted with the reference sequence Wuhan-1 (GenBank: NC 045512).

### 4.7. Protein Expression and Purification

The plasmid pαH encoding for the S protein ectodomain (residues 1–1208) of SARS-CoV-22 019-nCOV (GenBank: MN908947) stabilized in the prefusion conformation was kindly provided by Dr. Jason McLellan (the University of Texas at Austin—USA) [67]. Mutagenesis was conducted to create a HexaPro construct enabling the high-yield production of a stabilized prefusion spike protein [68]. The following substitutions are included at the HexaPro ectodomain: glycine at residue 614 (D614G), a “GSAS” substitution at the Furin cleavage site (residues 682–685), and proline at residues 817, 892, 899, 942, 986, and 987. The reference constructs and those incorporating minks’ mutations, detailed in Appendix A, were derived from HexaPro ectodomain through mutagenesis. All mutagenesis procedures followed the manufacturer’s instructions using the QuikChange Multi Site-Directed Mutagenesis Kit (Agilent, Santa Clara, CA, USA). The plasmids pcDNA3.4 TOPO encoding soluble and monomeric versions (residues 1–615) of human ACE-2 (NM_021804.3) and European mink ACE-2 (GenBank: QNC68911.1) were supplied by GeneArt company. Expression vectors for each construct were employed to transiently transfect FreeStyle 293F cells (Thermo Fisher Scientific, Waltham, MA, USA). Subsequently, the spike protein ectodomains and receptors were purified from filtered cell supernatants using Excel Ni-NTA columns (Cytiva, Marlborough, MA, USA), followed by size-exclusion chromatography on Superose 6 pG (Cytiva) or Superose 6 increase, and Superdex G200 H16/60 columns for spike proteins and receptors, respectively. Protein purity and integrity were assessed by SDS-PAGE and BluSafe staining under reducing conditions. Additionally, the proper folding of the spike protein in its prefusion conformation was evaluated by electron microscopy (Appendix A).

### 4.8. Surface Plasmon Resonance (SPR) Assays

All the SPR experiments were carried out in a Biacore X100 instrument using captured spikes as ligands and ACE2 receptors as analytes. The purified spike proteins were immobilized in protein A sensor chips (Cytiva) where the anti-Foldon human recombinant monoclonal antibody MF4 was previously coupled. First, the immobilization of the MF4 antibody was carried out at the reference and sample cells at approximately 4000 response units (RUs). Second, the immobilization of the spike proteins was performed at the sample cell at a level of approximately 700 RUs or approximately 300 RUs for assays involving mink or human ACE2 receptors, respectively. For the evaluation of SARS-CoV-2 adaptation to minks, the mink ACE2 receptor was injected at a concentration of 50 mM and flow rate of 40 μL/min for 100 s prolonging the dissociation time to 300 s. The binding report point estimated the binding response for each spike at 10 s before the end of receptor injection, the value that was normalized by the spike capture level. Statistical significance was calculated by one-way ANOVA, followed by Dunnett’s test with the B.1 spike mean as the control mean, using the GraphPad Prism 10.1.2 software. The binding kinetics/affinity analysis of mink spikes with the human ACE2 receptor was performed in a multicycle format assay injecting a range of 6 ACE2 concentrations (250–12.5 nM) at a flow rate of 40 μL/min. Association and dissociation phases were 138 s and 300 s, respectively. The binding data were fit to a 1:1 Langmuir binding model for the calculation of the kinetic parameters kon and koff. The KD was then calculated as the ratio of these two rate constants (koff/kon). In all the experiments, the regeneration of the chip was performed with 1 injection of a 30 mM HCl pH 1.5 solution prolonged to 60 s.

### 4.9. Structural Viewing

The PyMOL Molecular Graphics System v.2.5.7 was used to map the location of the ORF1ab and spike mutations of interest onto previously published SARS-CoV-2 structures of the NSP9 protein (PDB: 7BWQ) [38], RTC complex (PDB: 7CYQ) [36], and spike protein (PDB: 7BNN) [69].

## Figures and Tables

**Figure 1 ijms-25-05499-f001:**
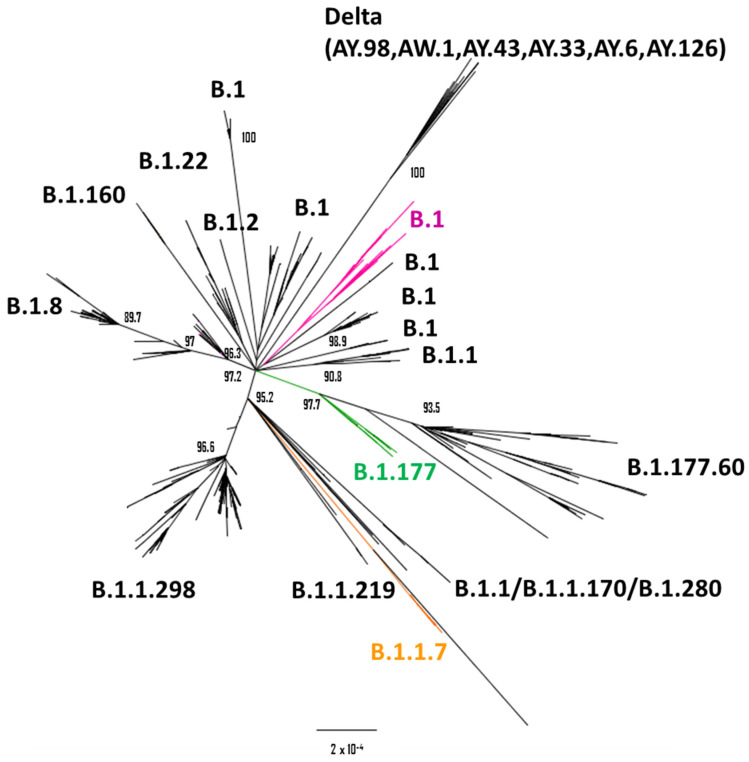
Phylogenetic tree based on the analysis of mink complete sequences published in GISAID (n = 1113 sequences). The clusters and sub-clusters defined by the Spanish sequences are shown in magenta, green, or orange for B.1, B.1.177, or B.1.1.7 lineages, respectively.

**Figure 2 ijms-25-05499-f002:**
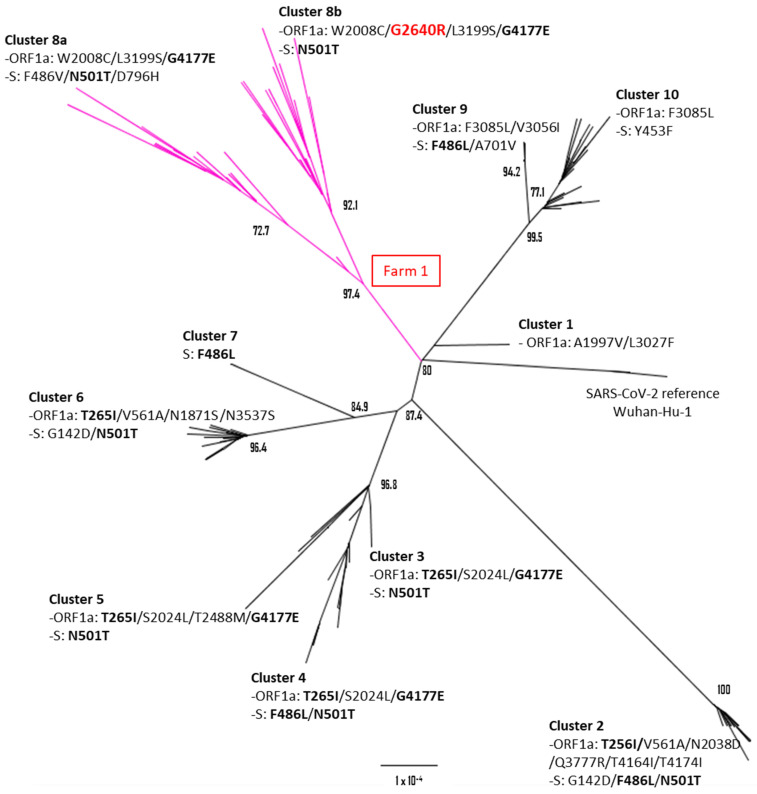
Phylogenetic tree based on the analysis of mink complete sequences published in GISAID (n = 233 sequences) belonging to the B.1 lineage. Numbers represent the percentage bootstrap support. The changes in ORF1a and spike which define each cluster are shown. Recurrent changes in four or more clusters are indicated in boldface. The clusters and sub-clusters defined by the Spanish sequences are shown in magenta and linked to the outbreak ID (box in red).

**Figure 3 ijms-25-05499-f003:**
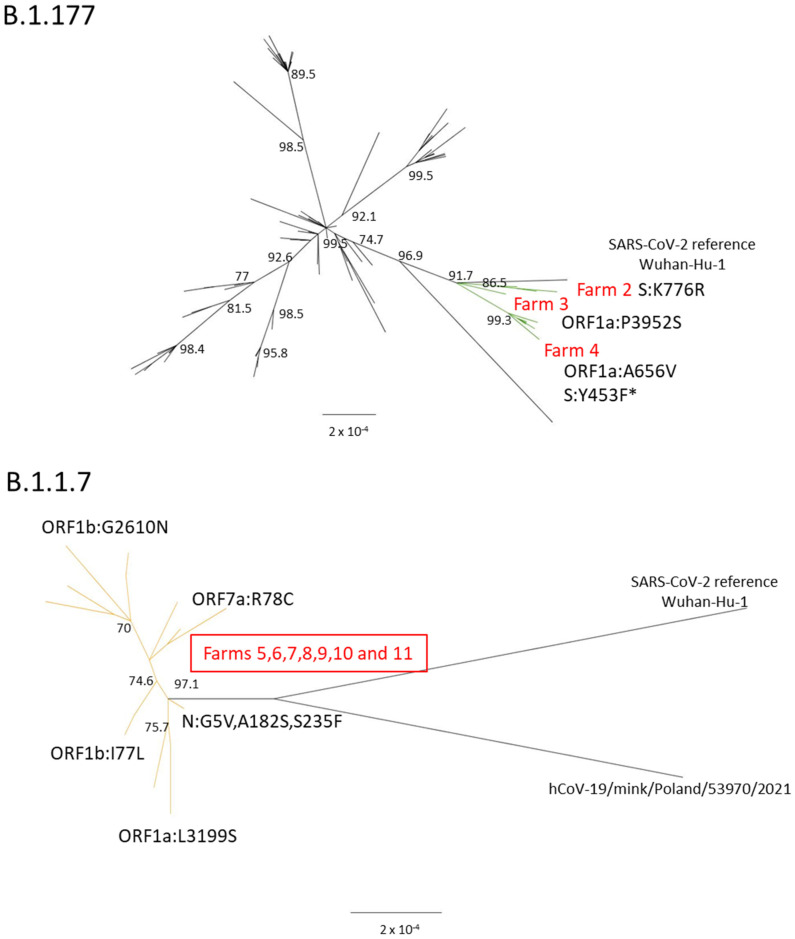
Phylogenetic trees based on the analysis of mink complete sequences belonging to the B.1.177 (**top panel**) and B.1.1.7 (**bottom panel**) lineages. * The mutation Y453F is only present in 2 of the sequences from the clade. The green and orange lines represent the clades where the Spanish sequences of SARS-CoV-2 are located.

**Figure 4 ijms-25-05499-f004:**
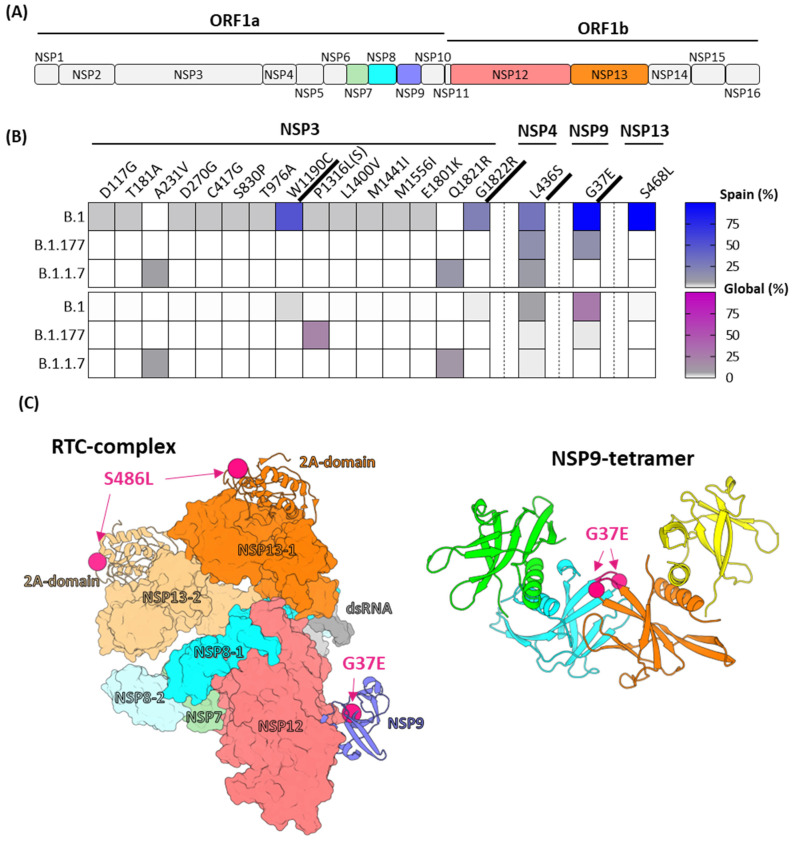
(**A**) Diagram of the SARS-CoV-2 ORF1ab polyprotein that includes 16 non-structural proteins (NSP). The NSPs comprising the RTC complex are highlighted in colors. (**B**) Heat map of mutations detected among Spanish mink sequences (top panel) or mink sequences available in GISAID database (bottom panel). (**C**) Localization of Spanish mink mutations in RTC complex. On the left: SARS-CoV-2 RTC complex (PDB: 7CYQ). Molecular surfaces of NSP7, NSP8, NSP12, and four domains of NSP13 are depicted. The alpha carbons of residues mutated are represented as hot pink spheres on the ribbon representation of NSP13 2A domain (S486L) and NSP9 (G37E). RTC components are color-coded according to the primary structure diagram. On the right: Localization of the G37E mutation in the NSP9 tetramer (PDB: 7BWQ). The four protomers of the tetramer are shown in ribbon structure with colors green, blue, orange, and yellow. The contact loop and the alpha carbons of the mutated residue (G37E) in mink sequences, located between the central protomers, are highlighted in hot pink.

**Figure 5 ijms-25-05499-f005:**
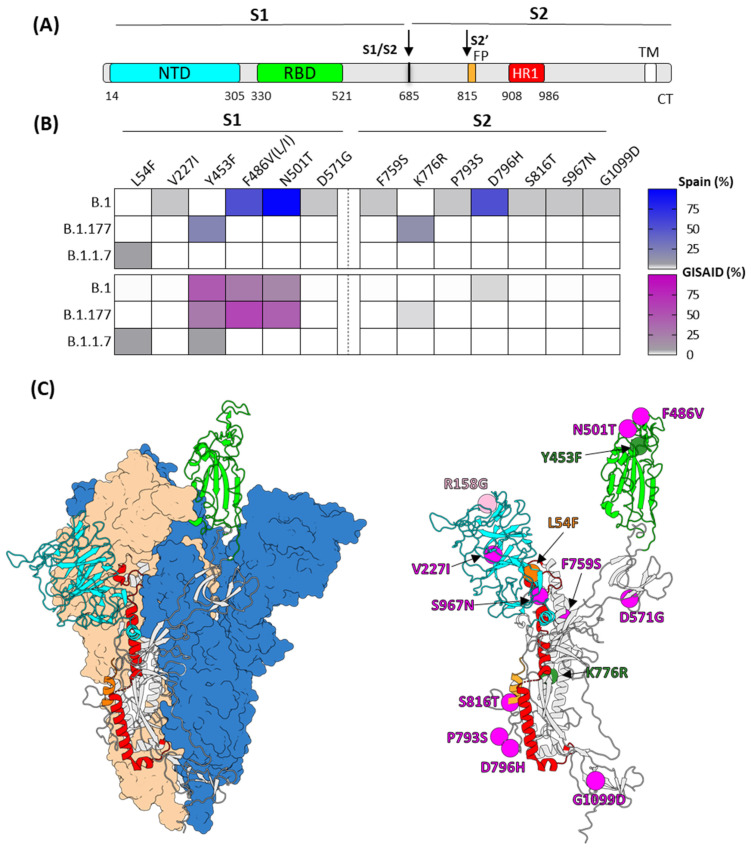
(**A**) Diagram of the SARS-CoV-2 spike protein primary structure, denoting the signal peptide (SP), N-terminal domain (NTD), receptor-binding domain (RBD), fusion peptide (FP), heptad repeat 1 (HR1), transmembrane domain (TM), and the cytoplasmatic tail (CT). The proteolytic cleavage sites that generate subunits S1/S2 and S2′ are indicated by arrows. (**B**) The heat map of mutations detected among the Spanish mink sequences (top panel) or mink sequences available in the GISAID database (bottom panel). (**C**) The localization of the Spanish mink mutations in the spike protein structure. On the left: SARS-CoV-2 spike trimer in its prefusion conformation with a single RBD in the RBD up-conformation (PDB ID:7BNN). The two RBD down protomers are shown as molecular surfaces in either blue or orange, and the RBD up protomer is shown in cartoon color as the diagram of the primary structure. On the right: the alpha carbon of the residues mutated in the Spanish mink sequences are shown in the RBD up protomer as spheres colored in magenta, green, orange, and pink, corresponding to mutations identified in B.1, B.1.177, B.1.1.7, and AY.98.1 lineages, respectively.

**Figure 6 ijms-25-05499-f006:**
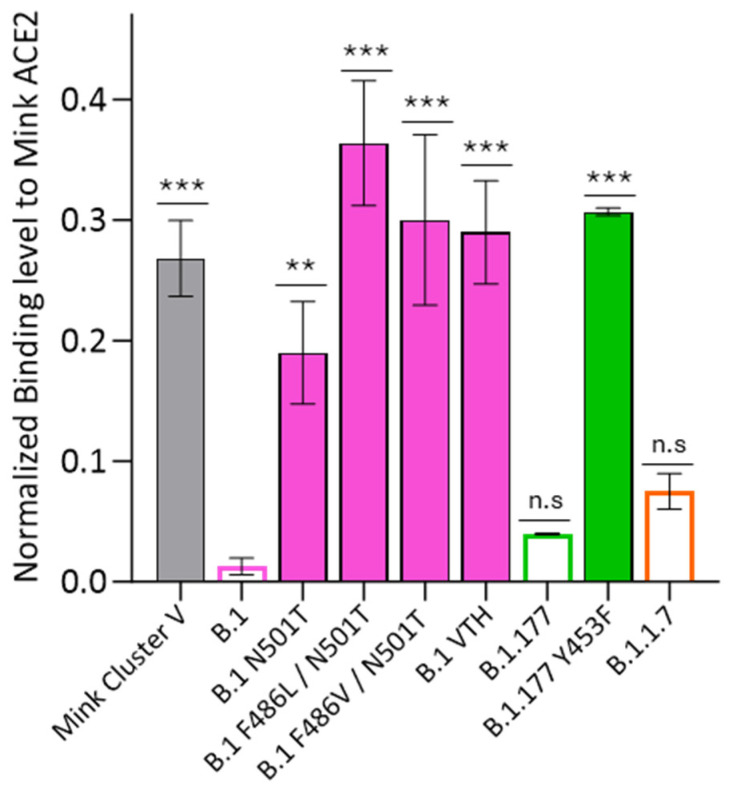
RBD mutations detected in minks strongly improve spike binding to mink ACE2 receptor. Binding response for SPR data was collected in triplicate, and the mean and the standard deviation are shown for normalized values to spike capture level. A Dunnett’s multiple comparisons test was performed for the mean of each spike with control mean (B.1 spike; ** *p* < 0.01, *** *p* < 0.001, n.s = not significant).

**Table 1 ijms-25-05499-t001:** Table describing the outbreak number, the date of outbreak notification to the national level, sample ID sequencing, and the GISAID data bank identification of the samples. NA: Not applicable.

Outbreak Number	Date of Notification to the National Level	Sample ID for Sequencing	Lineage	GISAID ID
1/2020	07/2020	R.001630/20-091	B.1	EPI_ISL_19096372
		R.001630/20-094	B.1	EPI_ISL_19096370
		R.001630/20-103	B.1	EPI_ISL_19096341
		R.001630/20-104	B.1	EPI_ISL_19096343
		R.001630/20-115	B.1	EPI_ISL_19096354
		R.001630/20-117	B.1	EPI_ISL_19096344
		R.001630/20-124	B.1	EPI_ISL_19096345
		R.001630/20-125	B.1	EPI_ISL_19096355
		R.001630/20-126	B.1	EPI_ISL_19096356
		R.001630/20-127	B.1	EPI_ISL_19096357
		R.001630/20-137	B.1	EPI_ISL_19096358
		R.001630/20-139	B.1	EPI_ISL_19096359
		R.001630/20-141	B.1	EPI_ISL_19096360
		R.001630/20-143	B.1	EPI_ISL_19096346
		R.001630/20-144	B.1	EPI_ISL_19096347
		R.001630/20-145	B.1	EPI_ISL_19096361
		R.001630/20-146	B.1	EPI_ISL_19096362
		R.001630/20-148	B.1	EPI_ISL_19096348
		R.001630/20-150	B.1	EPI_ISL_19096363
		R.001630/20-151	B.1	EPI_ISL_19096364
		R.001630/20-152	B.1	EPI_ISL_19096365
		R.001630/20-155	B.1	EPI_ISL_19096366
		R.001630/20-157	B.1	EPI_ISL_19096367
		R.001630/20-158	B.1	EPI_ISL_19096368
		R.001630/20-161	B.1	EPI_ISL_19096369
		R.001630/20-163	B.1	EPI_ISL_19096351
		R.001630/20-167	B.1	EPI_ISL_19096350
		R.001630/20-168	B.1	EPI_ISL_19096337
		R.001630/20-169	B.1	EPI_ISL_19096352
		R.001630/20-173	B.1	EPI_ISL_19096349
		R.001630/20-175	B.1	EPI_ISL_19096340
		R.001630/20-177	B.1	EPI_ISL_19096353
		R.001630/20-178	B.1	EPI_ISL_19096339
		R.001630/20-179	B.1	EPI_ISL_19096338
1/2021	21/01/2021	R.004000/20-08	B.1.177	EPI_ISL_19096333
		R.004000/20-15	B.1.177	EPI_ISL_19096334
		R.004000/20-16	B.1.177	EPI_ISL_19096335
2/2021	25/01/2021	R.000128/21-03	B.1.177	EPI_ISL_19096336
3/2021	17/03/2021	R.000776/21-17	B.1.177.50.1	EPI_ISL_19096332
		R.000776/21-18	B.1.177	EPI_ISL_19096331
		R.000871/21-58	B.1.177.50.1	EPI_ISL_19096330
		R.000871/21-62	B.1.177.50.1	EPI_ISL_19096329
		R.000871/21-68	B.1.177.50.1	EPI_ISL_19096328
		R.000871/21-97	B.1.177	EPI_ISL_19096326
		R.000871/21-114	B.1.177.50.1	EPI_ISL_19096327
		R.000871/21-116	B.1.177.50.1	EPI_ISL_19096325
		R.001042/21-78	B.1.177	EPI_ISL_19096324
4/2021	09/06/2021	R.001677/21-15	B.1.1.7	EPI_ISL_19096316
		R.001677/21-52	B.1.1.7	EPI_ISL_19096315
		R.001677/21-67	B.1.1.7	EPI_ISL_19096317
		R.2207/21-3	B.1.1.7	EPI_ISL_19096312
		R.2207/21-22	B.1.1.7	EPI_ISL_19096311
		R.2207/21-36	B.1.1.7	EPI_ISL_19096309
		R.2207/21-39	B.1.1.7	EPI_ISL_19096308
		R.2207/21-46	B.1.1.7	EPI_ISL_19096307
		R.0508/21-22	B.1.1.7	EPI_ISL_19096310
		R.0508/21-58	B.1.1.7	EPI_ISL_19096306
		R.1908/21-56	B.1.1.7	EPI_ISL_19096305
5/2021	25/06/2021	R.001738/21-19	B.1.1.7	EPI_ISL_19096313
6/2021	25/06/2021	R.001738/21-21	B.1.1.7	EPI_ISL_19096314
7/2021	25/06/2021	R.001807/21-59	B.1.1.7	EPI_ISL_19096318
10/2021	19/07/2021	R.002083/21-33	B.1.1.7	EPI_ISL_19096321
		R.002366/21-27	B.1.1.7	EPI_ISL_19096322
		R.002651/21-61	B.1.617.2	NA
		R.003234/21-26	B.1.617.2	NA
11/2021	19/07/2021	R.002026/21-100	B.1.1.7	EPI_ISL_19096319
		R.002026/21-101	B.1.1.7	EPI_ISL_19096320
16/2021	11/10/2021	R.002990/21-01	AY.98.1	EPI_ISL_19096323
17/2021	25/10/2021	R.003064/21-01	B.1.617.2	NA
		R.003064/21-33	B.1.617.2	NA

**Table 2 ijms-25-05499-t002:** Binding kinetic constants of mink spikes for human ACE2 receptor determined by surface plasmon resonance.

Mink Spikes	k_on_ (1/Ms) × 10^5^	k_off_ (1/s) × 10^−2^	KD (nM)
Mink Cluster V	0.91 ± 0.117	0.11 ± 0.009	12.3 ± 1.6
B.1	0.89 ± 0.041	0.92 ± 0.031	103.8 ± 8.3
B.1-N501T	1.05 ± 0.008	0.25 ± 0.019	24.2 ± 1.6
B.1-F486L/N501T	0.8 ± 0.007	0.94 ± 0.091	117.8 ± 10.6
B.1-F486V/N501T	0.79 ± 0.02	0.69 ± 0.023	87.0 ± 5.1
B.1-VTH	0.69 ± 0.027	0.79 ± 0.117	114.5 ± 12.4
B.1.177	0.89 ± 0.101	1.02 ± 0.042	114.4 ± 8.2
B.1.177-Y453F	0.91 ± 0.008	0.1 ± 0.003	11.4 ± 0.5
B.1.1.7	1.6 ± 0.1	0.2 ± 0.019	12.3 ± 0.4

## Data Availability

All the sequences are available in GISAID.

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
