# Peer review of "Genomic Context of SARS-CoV-2 Outbreaks in Farmed Mink in Spain during Pandemic: Unveiling Host Adaptation Mechanisms"

_ijms, 2024, doi:10.3390/ijms25105499_

Round 1

Reviewer 1 Report

Comments and Suggestions for Authors

The manuscript reported the analysis of the genome of SARS-CoV-2 strains derived from outbreaks in farmed minks in Spain during 2020-2021. It did not highlight the novelty/interest of the results for the scientific community and the significance, if any, for the evolutionary mechanism of the virus of the viral fitness.

Comments on the Quality of English Language

English language could be improved avoiding repetitions and typing errors.

Author Response

Genomic context of SARS-CoV-2 in farmed mink outbreaks in Spain during pandemic: unveiling host adaptation mechanisms.

Response to Reviewer 1 Comments

1. Summary

Thank you very much for taking the time to correct the text. We have made efforts to enhance the text by addressing language and punctuation errors. Based on your feedback, we believe that the value of this work is particularly evident between lines 467 and 475. In these lines, we meticulously describe how analyzing mutations present in the complete genome, not just the spike protein, along with their phenotypic characteristics, has improved decision-making in virological surveillance. Additionally, we emphasize the importance of exploring other regions of the SARS-CoV-2 genome beyond the spike protein, which are typically overlooked, thereby introducing new insights to the field.

Moreover, the collaboration between veterinary and human viral surveillance agencies, as demonstrated in this work, has contributed to our understanding of SARS-CoV-2 viral evolution and its adaptation to a host that is known to be vulnerable to coronaviruses as well as other highly significant respiratory viruses.

Reviewer 2 Report

Comments and Suggestions for Authors

In the manuscript by Iglesias-Caballero et al., the authors analyzed SARS-CoV-2 whole genome sequences in minks at farms sites where outbreaks occurred.  They placed the sequenced genomes into distinct lineages and observed that mutations from consensus did not define new lineages.  Rather, these mutations were cumulative during the outbreaks, and the groups of mutations defined lineages.  They identified proteins that harbored the majority of the mutations and performed binding analysis for the Spike protein with mink ACE2.  Overall, the results are thoroughly presented and support the conclusions of the authors.  I have only minor suggestions for revision to improve the clarity of the manuscript.

Table 1 – The format for presentation of dates should be shown. Is it DD/MM/YYYY?  The bottom of column 1 shows months 16 and 17?  What is the purpose of showing sample ID number?  Is this helpful to the reader?  The GISAID Ids are missing.

Line 114 – Spanish should be capitalized.

Figure 6 – these data need to be properly analysed with statistical analyses performed.  Only then are the authors’ conclusions fully supported.

Author Response

Genomic context of SARS-CoV-2 in farmed mink outbreaks in Spain during pandemic: unveiling host adaptation mechanisms.

Response to Reviewer 2 Comments

1. Summary

Thank you very much for your time in correcting the text. We also appreciate the overall assessment of the article and look forward to addressing your comments appropriately. Please find the detailed responses below and the corresponding corrections highlighted in the re-submitted files.

2. Questions for General Evaluation

Reviewer’s Evaluation

Response and Revisions

Does the introduction provide sufficient background and include all relevant references?

Yes

Are all the cited references relevant to the research?

Yes

Is the research design appropriate?

Yes

Are the methods adequately described?

Yes

Are the results clearly presented?

Yes

Are the conclusions supported by the results?

Yes

3. Point-by-point response to Comments and Suggestions for Authors

Comments 1: Table 1 – The format for presentation of dates should be shown. Is it DD/MM/YYYY?  The bottom of column 1 shows months 16 and 17?  What is the purpose of showing sample ID number?  Is this helpful to the reader?  The GISAID Ids are missing.

Response 1: Thank you for pointing this out. We would like to clarify that the first column is not the date; it's the way to number the outbreaks. The date in the format DD/MM/YYYY is located in the following column. The idea of ​​showing the sample ID is for it to be associated with the GISAID ID and for the phylogenetic analyses to be replicable for the rest of the scientific community. The GISAID IDs have already been added.

Comments 2:
Line 114 – Spanish should be capitalized.

Response 2: Thank you for noticing it. Corrected in the text (lines 116,122)

Comments 3: Figure 6 – these data need to be properly analysed with statistical analyses performed.  Only then are the authors’ conclusions fully supported.

Response 2: Thank you for pointing this out. Figure 6 has been now updated to incorporate statistical significance analyses using one-way ANOVA followed by Dunnett’s test. The increased binding of mink spikes to the mink ACE-2 receptor compared to the B.1 spike is robustly supported by these statistical analyses. The results reveal a significant difference with P < 0.001 for cases containing the Y453F mutation (Cluster V and B.1.177-Y453F) or the double mutations at residues 486 and 501 (B.1 N501T/F486L, B.1-N501T/F486V, and B.1-VTH), and P < 0.01 for B.1 N501T.

Details of the statistical analyses performed are now indicated in the Materials and Methods section (lines 611-613) “Statistical significance was calculated by one-way ANOVA, followed by Dunnett’s test with the B.1 spike mean as the control mean, using GraphPad Prism 10.1.2 software”; and in Figure 6 legend (lines 293-295) “A Dunnett’s multiple comparisons test was performed for the mean of each spike with control mean (B.1 spike; **P < 0.01,***P < 0.001, n.s = not significant)”.

.

Reviewer 3 Report

Comments and Suggestions for Authors

In the present study, the authors analyzed whole genome sequences and some spike sequences from 18 outbreaks of SARS-CoV-2 and identified four distinct lineages. They found numerous not lineage-defining mutations and found that outbreaks were predominantly associated with different groups of mutations rather than specific lineages. This study should be interesting to readers.

1. Abstract section needs to be rewritten to highlight why this research is being conducted and its importance.

2. P values are missing in Figure 6.

3.  Are there any evidence to show that spike protein mutations detected here may enhance or decrease SARS-CoV-2 adaptability?

4. In method section, please provide more information how the samples are collected and sequenced.

5. Please provide more information of the primers used for qRT-PCR.

Author Response

Genomic context of SARS-CoV-2 in farmed mink outbreaks in Spain during pandemic: unveiling host adaptation mechanisms.

Response to Reviewer 3 Comments

1. Summary

Thank you very much for your time in correcting the text. We also appreciate that you find the work interesting. Please find the detailed responses below and the corresponding corrections highlighted in the re-submitted files.

2. Questions for General Evaluation

Reviewer’s Evaluation

Response and Revisions

Does the introduction provide sufficient background and include all relevant references?

Can be improved

Are all the cited references relevant to the research?

Can be improved

Is the research design appropriate?

Yes

Are the methods adequately described?

Can be improved

Are the results clearly presented?

Can be improved

Are the conclusions supported by the results?

Can be improved

3. Point-by-point response to Comments and Suggestions for Authors

Comments 1: Abstract section needs to be rewritten to highlight why this research is being conducted and its importance.

Response 1: Thank you for the suggestion. Given the relevance of the issue pointed out, but due to space constraints, we have added a sentence in line 16 that highlights the importance of monitoring these mutations due to their potential impact on human transmission “The potential risk of transmission to humans raises crucial questions about mutation accumulation and its impact on viral fitness”.

Comments 2: P values are missing in Figure 6.

Response 2: Thank you for pointing this out. Figure 6 has been now updated to incorporate statistical significance analyses using one-way ANOVA followed by Dunnett’s test. The increased binding of mink spikes to the mink ACE-2 receptor compared to the B.1 spike is robustly supported by these statistical analyses. The results reveal a significant difference with P < 0.001 for cases containing the Y453F mutation (Cluster V and B.1.177-Y453F) or the double mutations at residues 486 and 501 (B.1 N501T/F486L, B.1-N501T/F486V, and B.1-VTH), and P < 0.01 for B.1 N501T.

Details of the statistical analyses performed are now indicated in the Materials and Methods section (lines 611-613) “Statistical significance was calculated by one-way ANOVA, followed by Dunnett’s test with the B.1 spike mean as the control mean, using GraphPad Prism 10.1.2 software”; and in Figure 6 legend (lines 293-295) “A Dunnett’s multiple comparisons test was performed for the mean of each spike with control mean (B.1 spike; **P < 0.01,***P < 0.001, n.s = not significant)”.

Comments 3: Are there any evidence to show that spike protein mutations detected here may enhance or decrease SARS-CoV-2 adaptability?

Response 3: We believe that the mutations detected here, specifically those at positions 453 (Y453F), 486 (F486V or F486L), and 501 (N501T), enhance SARS-CoV-2 adaptability to minks (lineS 414-416). Evolutionary improvements in spike protein attachment to human cells, indicated by stronger affinity to the human receptor ACE2, have been observed since the beginning of the pandemic, exemplified by variants like Delta or Omicron (ref 1). The finding that the mink spike's ability to bind to the mink cell receptor is ≥10 times higher compared to human spikes suggests a similar behavior that facilitates greater adaptation to the mink host.

Comments 4: In method section, please provide more information how the samples are collected and sequenced.

Response 4: Thank you for your comment. I believe that in section 4.1 (between lines 476 and 484), both the distribution of the samples used and the method by which they are taken are explained. In section 4.3 (between lines 500 and 526), there is a comprehensive description of all the steps from genome amplification to the reagents used for library preparation covering the whole process.

Comments 5: Please provide more information of the primers used for qRT-PCR.

Response 5: Thank you for your comment. In this paper, we only describe in the text the primers designed in the laboratory, as is the case with the spike sequencing primers between lines 512 and 515. The qRT-PCR primers were those described and designed in the works cited in references 48 and 49, where their description can be found. In the text, we clarify their targets and the reasons for their use, but since they are not the result of our work and to economize the text size, we believe it is better to reference them in their citation.

Round 2

Reviewer 1 Report

Comments and Suggestions for Authors

The abstract was modified with one sentence bringing a speculative comment, but the Discussion was unmodified and thus the criticisms are still present about the novelty/utility/benefit for the scientific community derived from the manuscript. In other words, please, explain the significance of your study.

Author Response

Genomic context of SARS-CoV-2 in farmed mink outbreaks in Spain during pandemic: unveiling host adaptation mechanisms.

Response to Reviewer 1 Comments

1. Summary

Thank you again for your suggestions. To adequately respond to your proposal, we have included two additional paragraphs this time.

·         “To face this risk, the association of animal and human health laboratories in this work, under the One Health umbrella, enables the characterization of mink viruses with all the technology developed during pandemic for the analysis of human SARS-CoV-2, enhancing virus surveillance by moving beyond mere sequence description in mink’s coronaviruses.” (lines 323-327)

·         “This work demonstrates the potential of the development of methodology in the surveillance of human SARS-CoV-2 viruses could be used for monitoring animal coronaviruses. Our study highlights the importance of conducting phylogenies with complete genomes, as it underscores the significance of characterizing the entire genome for tracing outbreaks or mechanisms of host adaptation. Additionally, this work illustrates the importance of phenotypic virus studies, not considering only the hypothetical impact of genetic changes, showing that only through the combination of genomic and functional/antigenic methods can surveillance networks propose appropriate responses to public health emergencies.” (lines 479-486).

We believe that these new paragraphs along with the previously presented parts (listed below) express the benefit for the scientific community of this work.

·         “Surprisingly, the sequences did not group according to their respective lineages, as typically observed in viral phylogenetic trees. Instead, the sequences clustered together based on the specific outbreaks they were associated with. This clustering pattern by outbreak is significant and may be attributed to the rapid accumulation of mutations, particularly in the ORF1a polyprotein and in the spike protein” (lines 352-356)

·         “Moreover, our study provided evidence of adaptive mechanisms based on an improvement of the spike binding to mink ACE2, the cellular receptor used by SARS-CoV-2 to enter host cells. Specifically, we identified host markers related to changes at RBD positions 453, 486 and 501. These mutations could potentially enhance the affinity of the spike protein for the mink ACE2 receptor, facilitating viral entry and replication in minks” (lines 415-420)

·         “Furthermore, analyzing the polyprotein in conjunction with the spike can provide a more thorough understanding of host markers and viral transmission in the community, there-by enhancing our knowledge of various outbreaks.” (lines 479-481)
